# Understanding older adults' perception, acceptance, and adoption of smart home technologies

Pireh Pirzada[1], Adriana Wilde[2,3]*, Gayle H. Doherty[4], David Harris-Birtill[5]

1 School of Electronic Engineering and Computer Science, Queen Mary University of London, London, United Kingdom, 2 Digital Health and Biomedical Engineering Research Group, School of Electronics and Computer Science, University of Southampton, Southampton, United Kingdom, 3 Centre for Health Technologies, University of Southampton, Southampton, United Kingdom, 4 School of Psychology and Neuroscience, University of St Andrews, St Andrews, United Kingdom, 5 School of Computer Science, University of St Andrews, St Andrews, United Kingdom

* a.wilde@soton.ac.uk

## Abstract

Smart home technologies have increased in popularity and affordability in recent years, however, there is limited research on their adoption specifically among older adults. This study aims to uncover incentives and barriers to the adoption of smart home components, given the impact of COVID-19 on older adults' quality of life (QoL). For this purpose, online audio-recorded interviews were conducted with 21 participants aged between 65 and 90 years from the UK, Malta, and Pakistan. Participants were shown various smart home technology images and videos and asked for their perception, existing knowledge, and current use. The corresponding audio recordings were transcribed and thematically analysed. Findings revealed that older adults experienced a decline in QoL during COVID-19, accompanied by an increased reliance on digital technologies. While participants recognised the potential of smart home devices to enhance independence and well-being, their knowledge of available options was limited. Key barriers included affordability, privacy and trust concerns, compatibility and integration issues, and environmental and social considerations. Cross-cultural differences emerged, with European participants reporting higher familiarity than those in Pakistan, particularly rural areas where availability was limited. These results highlight the need for designers and policymakers to improve affordability, usability, and cultural adaptability in order to support older adults' acceptance and meaningful adoption of smart home technologies.

## Introduction

The global population of older adults (aged 65 or above) is increasing due to improved health services and life expectancy [1], which, compounded with a decline in fertility rate, has created a demographic shift. The upsurge in the ageing population

**Data availability statement:** All relevant data cannot be shared publicly because of sensitive information, ethical restrictions and participant confidentiality. Summarised and anonymized data are available within the paper and its Supporting information, in accordance with the ethics approval granted by the University of St Andrews (approval code: CS14204).

**Funding:** This study was financially supported by the University of St Andrews in the form of PhD funding awarded to PP. This study received further support from the University of St. Andrews School of Computer Science in the form of interview equipment and participant vouchers. No additional external funding was received for this study. The funders had no role in study design, data collection and analysis, decision to publish, or preparation of the manuscript.

**Competing interests:** The authors have declared that no competing interests exist.

impacts public and private health care systems [1] since an increased life expectancy is associated with decreased health. Ageing increases the risk of non-communicable disease; thus, older adults are at higher risk of having one or more health conditions, such as cardiovascular disease, dementia and other neuro-degenerative conditions [2]. This can result in high costs for families, caretakers, and public and private health care [3]. To provide support for the ageing population and maintain their quality of life (QoL), various smart home technologies are now available on the market, with many more under research and development.

Smart homes are environments with various devices interacting with each other to automate different operations within a residential setting. The capabilities of smart home technologies provide the older population with a variety of ways to monitor and maintain their health and wellbeing, achieving the goal of remaining independent for a longer time. This is possible thanks to the amalgamation of sensors and features of smart home technologies, which provide various benefits to support the needs of older adults, such as daily task automation, health, activity, and emergency monitoring, as well as social connectivity [4]. These technologies aim to improve QoL, understood not only as the absence of disease but as the overall mental, physical, and social well-being of older adults [5].

The COVID-19 pandemic underscored the vulnerability of older adults, particularly those over 80, and reinforced the need for remote healthcare monitoring systems integrated into smart homes. Such systems may help reduce the mental and financial pressure on healthcare services, families, and caretakers by enabling older adults to remain independent while maintaining safety [4,6–8].

While previous studies have identified positive attitudes toward smart home technologies and barriers such as privacy, trust, and affordability, most research has been conducted in western contexts and rarely compared cross-cultural perspectives [4]. For example, a study from the United States show that willingness to adopt smart homes, assistive robots, and other emerging technologies is shaped by health, income, and prior technology experience [9], while research in East Asia (Taiwan) [10] highlights the importance of family and social influence in shaping older adults' adoption of eHealth services. However, we found no studies that directly compare older adults' adoption patterns across regions, cultures, or socioeconomic contexts (e.g., income, household structure, rural–urban differences), leaving a gap in understanding how cultural norms and structural factors jointly shape technology adoption.

In addition, limited work has explored adoption in the context of COVID-19, despite its significant impact on older adults' quality of life [4]. This study addresses these gaps by exploring older adults' perceptions, awareness, and adoption of smart home technologies in the UK, Malta, and Pakistan. Using thematic analysis of online interviews, it identifies incentives and barriers to adoption. It explores both well-established concerns (e.g., privacy, trust, affordability) and novel factors, and examines how these factors influence quality of life. Building on our previous work, this paper contributes new insights into cross-cultural differences in technology adoption and highlights factors that may improve smart home acceptance in diverse contexts.

## Motivation and objectives

The main objective of this study was to understand the current perception, awareness and adoption rate of smart home technologies among older adults. Notably, existing literature offers limited insight into how commercially available smart home technologies are perceived and used by older adults, and to the best of our knowledge, no recent research has specifically focused on this demographic's engagement with these solutions.

To reduce mental and financial pressure on healthcare services and families, various technologies continue to be researched and developed so that older adults may be supported in leading independent lives without compromising their health and safety. However, despite the continuous advances in research and development and the growing availability of products on the market, questions remain regarding technology interaction, acceptance and adoption. This could be due to several factors which may impact the acceptance and adoption of assistive technologies. Our review identified some of the better-known factors, such as system design, ethical concerns, user experience, awareness about the technology, and personalisation [4]. This paper investigates how older adults perceive, understand, and adopt commercially available smart home technologies, addressing a gap in the existing research where comprehensive insights are currently lacking.

Prior studies report positive attitudes toward smart technologies [11–16], however, adoption remains cautious, due to concerns over reliability, security, privacy, and trust [17–24]. For example, individuals expressed preference for in-person healthcare appointments rather than online [11]. Yet, during the pandemic, people became more accepting of communication technology [25], and therefore this reported reticence may continue to decrease over time. Attitudes to other smart technologies may have changed as well, especially as more products are now available on the market. Hence, it is important to explore attitudes towards smart home technologies, including the reasons behind older adults' resistance. Insights from this study may help identify gaps to potentially improve the acceptability and adoption of smart home technologies among older adults.

## Methodology

This research was granted ethical approval at the University of St Andrews (approval code: CS14204). Written informed consent was obtained electronically through Qualtrics before participation. All participants were adults aged 65 and above, age threshold chosen to be consistent with definitions of older adults in technology-adoption research [26]. Participants were recruited between 28.05.2019 to 15.12.2021. This section provides details about the participants, the interviews conducted, and the data collection process.

### Interviews

Online audio-recorded interviews were conducted via Microsoft Teams in the participants native language including English, Urdu, or Sindhi with 21 participants aged 65–85. The observed age range reflects recruitment and participation rather than predefined exclusion criteria. Participants were shown and described images and videos of smart home technologies that are commercially available (Table 2 and Table 3 in [4]) and asked about their opinions, perceptions, existing knowledge and current use of these technologies. Initially, advertisements were titled "Smart homes for the elderly to promote health and well-being interviews", with accompanying images of older adults which was later changed to "Smart homes for health and well-being interviews" as the focus was on the technology rather than the frailty or vulnerability of participants. The images used were also changed to show examples of smart home technology which resulted in increased recruitment of participants. These characteristics were thought to impact their willingness to participate [27] which this paper discusses in the Discussion section.

The study was open to participants from different parts of the world to understand any cultural, social, and economic differences that could affect how they perceive and use smart technologies. Participation reflects uptake rather than targeted sampling. This approach aimed to explore the impact of international variation on the generalisability of the results,

addressing a gap in current research on cross-cultural influences. Thematic analysis was used to examine qualitative data from participant interviews, systematically identifying and interpreting recurring patterns [28]. By coding the transcripts, similar responses were grouped into key themes which were central to participants' experiences with smart home technologies which are listed in this paper in the Results and findings section. Interview transcripts were analysed using Braun and Clarke's six-phase framework for thematic analysis [29]. Coding was conducted manually on transcripts, using an inductive approach at the semantic level, supplemented by elements of latent analysis to capture contextual factors. An audit trail of coding decisions and theme refinement was maintained to ensure transparency.

Participants experienced technical issues during the interviews (e.g., problems with connection, audio, video controls, and screen-sharing). The interviews, on average, were 45 minutes long and were audio-recorded, transcribed, and analysed for themes [28]. Recruitment methods included social media, flyers, emails, digital magazines, and word of mouth. Participants contacted the researcher via email in response to the recruitment advertisement and to schedule interviews. Some participants stated that some of their peers did not have enough technical skills and struggled using technology, preventing participation. Therefore, the interviews conducted only reflect the knowledge and perception of those participants who were already relatively competent in using digital platforms. Also, as participants have to volunteer to take part in the study, it is possible that the participants are a self-selecting group who are more likely to be sociable and outgoing compared to the general population. Some participants also mentioned that they were initially reluctant to sign up as they thought they might not know enough to answer technology-related questions. To mitigate this, unfamiliar smart home technologies were explained during interviews using images and videos, enabling participants with lower prior knowledge to engage meaningfully.

A token of appreciation (a shopping voucher for GBP 10) was given to UK participants. Providing incentives to international participants was not covered by ethics approval and was not feasible within the approved logistical and administrative arrangements. The researcher briefed participants about the interview process, allowing them to skip questions if desired, but all participants responded to every question and willingly shared additional information. Unfamiliar smart home technologies were explained during the interviews to ensure participants' understanding.

### Eligibility

- Inclusion criteria: aged ≥65, able to communicate in English/Urdu/Sindhi, internet access.

- Exclusion criteria: under 65, unable to participate due to no internet access, or incomplete/withdrawn interview.

All 21 respondents met the criteria and were included in the study.

### Interview format

The interview was structured in three phases, focusing on participants' knowledge, perceptions, and current and future adoption of smart home technologies. The first covered personal, social, and COVID-19 impact on the participant's QoL. The second phase entailed a presentation containing images and website links to smart home objects such as heating, ventilation, air conditioning (HVAC), Mixed Reality (AR or VR), Google Jacquard and others in Table 1 [4].

Participants were then shown some videos of commercially available smart home technologies listed in use in Table 2 [4]. Finally, in the third phase, a questionnaire on Qualtrics elicited participants' reasons for hesitation to adopt various smart home technologies. All three interview phases were conducted within a single session, with interviews lasting approximately 45 minutes on average. The questionnaire is shown in Supporting information S1 Appendix.

The smart home technologies in this study were selected based on market research for their ability to enhance the safety and independence of older adults. The technologies in Tables 1 and 2 offer essential features for activities of daily life including health monitoring, and home automation. Table 2 lists fully-fledged, commercially available smart home

**Table 1. Commercial products categorised [4].**

| Category | Description |
|---|---|
| Smart Pillbox[a] | A pillbox is a component of a smart device that connects to smartphones or laptops, offering reminders and dispensing pills on a set schedule. |
| Smart Door[b] | A smart door allows users to control the lock through various methods, including a smartphone, thumbprint, pin code, voice recognition, and more. |
| Security and Activity Monitoring Cameras[c] | Cameras allow users to monitor home security and track the activities of older adults. |
| Smart Heating, Ventilation, Air conditioning System[d] | Smart heating, ventilation, and air conditioning systems offer temperature control, save energy, and can be controlled via a smartphone. |
| Personal Emergency Response Systems[e] | A personal health monitoring system can be used to call for help in emergencies. Such as wearable devices like push-button necklaces, watches, and belts. |
| Smart Clothes | Monitoring location using GPS, detecting falls, and tracking physiological health, etc. |
| VR/AR | Virtual, augmented, or mixed reality for connecting with people, attending meetings, or engaging in leisure activities. |
| Other | Monitoring blood glucose, blood pressure, heart rate, blood oxygen levels, and any other smart systems to track overall well-being and ADL. |

[a]Pillbox, https://www.tricella.com/smart-pillbox

[b]Smart Door, https://nuki.io/en/smart-door/

[c]Arlo, https://www.arlo.com/uk/use-cases/assisted-living/

[d]Fibaro, https://www.fibaro.com/en/smart-home-in-use/smart-hvac

[e]Lifeline, https://www.lifeline24.co.uk/

systems that provide comprehensive solutions, integrating multiple assistive features. This was to assess participants' awareness, current adoption rates, and potential barriers for older adult users. Understanding these challenges ensures that future studies can address them when designing these technologies. These findings are directly applicable to real-world scenarios, providing insights into improving the QoL for older adults.

## Participant information

The participant information sheet was presented, and consent was collected using Qualtrics. Selection criteria included those who were able to consent for themselves and had no diagnosed cognitive impairment such as dementia. All participants were living independently, thus not dependent on a carer. Participants were provided with contact details of the researchers and Age UK's advice line in case they had concerns or felt distressed by issues raised during the interview process. Participants were informed they could withdraw at any time before data anonymisation, after which withdrawal was not possible. Researchers contact details were provided for this purpose.

Participants reported no difficulty when performing activities of daily life, although many felt that their memory had declined with age. Participants also had at least one underlying health condition, including chronic pain, back problems, thyroid, diabetes, cardiovascular and mental health issues. These were managed with medication and medical devices (blood pressure monitors, thermometers and continuous glucose monitoring systems, e.g., Freestyle Libre). Table 3 shows demographics, including division by urban and rural areas (where health facilities, electricity, transport, and other resources are limited).

**Table 2. Smart home technology products and services shown to participants. Table footnotes include links to video demonstrations presented during the interviews [4].**

| Product | Description |
|---|---|
| Lively[a] | Lively is a smart home monitoring system that tracks daily activities using environmental sensors. Participants were shown a video demonstrating its use case. |
| Geeny[b] | Geeny offers various sensors and alarms for monitoring health and well-being at home. It provides smart plugs, environmental sensors, emergency push buttons, and more, all of which can be configured within a home. A use case video was shown to participants and explained that they could buy these individual components and install them within their homes. |
| Just checking[c] | Just checking allows monitoring the activities of residents using sensors that can be attached to doors, walls, and other objects within the house. |
| Canary care[d] | Canary care enables users to monitor activities within the home and provides detailed reports that can be viewed on a smartphone or laptop. It includes environmental sensors to track bathroom visits, daily movement, sleep patterns, and medication reminders. |

[a]Lively, https://www.youtube.com/watch?v=7KVHMlHLZr0

[b]Geeny, https://www.youtube.com/watch?v=BCK-v0cWE3k

[c]Just checking, https://www.youtube.com/watch?v=sxdcRwRTIr0

[d]CanaryCare, https://www.youtube.com/watch?v=jPmRgLGdQd8

**Table 3. Demographic distribution (n=number of participants).**

| Age Group | | Country | | Gender | |
|---|---|---|---|---|---|
| *Age Group* | *n* | *Country* | *n* | *Gender* | *n* |
| 65 - 70 | 10 | United Kingdom | 13 | Female | 9 |
| 71 - 75 | 6 | Malta | 1 | Male | 12 |
| 76 - 80 | 4 | Pakistan (Urban) | 3 | – | – |
| 81 - 90 | 1 | Pakistan (Rural) | 4 | – | – |

Participants' current technology adoption is shown in Table 4. All participants from rural areas (villages) of Pakistan were living with extended family members which could reflect cultural norms. Most European respondents lived alone or with their partners. The exception was a UK participant living in a multi-generational household. Table 4 also shows that laptops/PCs and smartphones were widely used among participants, with the highest usage observed in the UK, likely due to the higher number of participants from that region. However, the adoption of smart medical devices was low, with only one UK participant using such technology, suggesting limited access or a lack of perceived benefit among older adults. Similarly, smart home components were used exclusively in Europe but by only a small number of participants, indicating slow adoption rates in this demographic.

Participants from rural Pakistan were recruited by word of mouth with local assistance given where needed from their family members. Consent forms and other media were translated into the appropriate language. Table 3 also shows participant division by rural and urban areas. This distinction is important to show that participants' opinions were collected from privileged (urban) and underprivileged (rural) areas. Rural areas represent areas where health facilities, electricity, transport, and other resources are limited. An example image of the rural area is shown in Fig 1. These images are obtained with permission from the residents where faces of people have been removed to keep data anonymous.

**Table 4. Participants' living arrangements and technology adoption counts.**

| Participant Technology Adoption | |
|---|---|
| **Living alone (All countries)** | **5** |
| — in Malta | 1 |
| — in the UK | 4 |
| **Living with a partner (All countries)** | **10** |
| — in Pakistan | 2 |
| — in the UK | 8 |
| **Living in an extended family (4 + people) (All countries)** | **6** |
| — in Pakistan (Rural) | 4 |
| — in Pakistan (Urban) | 1 |
| — in the UK | 1 |
| **Users of a laptop/PC (All countries)** | **17** |
| — in Pakistan (Urban) | 3 |
| — in Malta | 1 |
| — in the UK | 13 |
| **Users of a smartphone (All countries)** | **17** |
| — in Pakistan (Urban) | 3 |
| — in Malta | 1 |
| — in the UK | 13 |
| **Users of smart medical devices (only UK)** | **1** |
| **Users of social media (All countries)** | **17** |
| — in Pakistan (Urban) | 3 |
| — in Malta | 1 |
| — in the UK | 10 |
| **Users of a communication media (All countries)** | **18** |
| — in Pakistan (Urban) | 3 |
| — in Malta | 1 |
| — in the UK | 13 |
| **Users of smart home components (Europe only)** | **3** |
| **Total** | **21** |

## Results and findings

### Quality of life (QoL)

All participants agreed with the definitions of QoL and the conceptual diagram shared [4,5], reporting satisfaction with their QoL. Some added spiritual and environmental aspects to the definition, referring to Maslow's hierarchy [30], prioritising basic needs before progressing to higher levels. Five participants (24%) shared that maintaining good health in inadequate physical environments among those unable to afford quality housing and healthcare, especially in developing countries, can be quite challenging. European participants tended to emphasise mental health and autonomy in their QoL definitions, while Pakistani participants (particularly rural) highlighted financial stability and family support. This aligns with collectivist versus individualist orientations described in existing framework [31,32], where our findings suggest that family-based interdependence is strongly valued in Pakistan. Overall, Health was reported as a primary need, whereas others also included environment, housing, food, and financial stability as important.

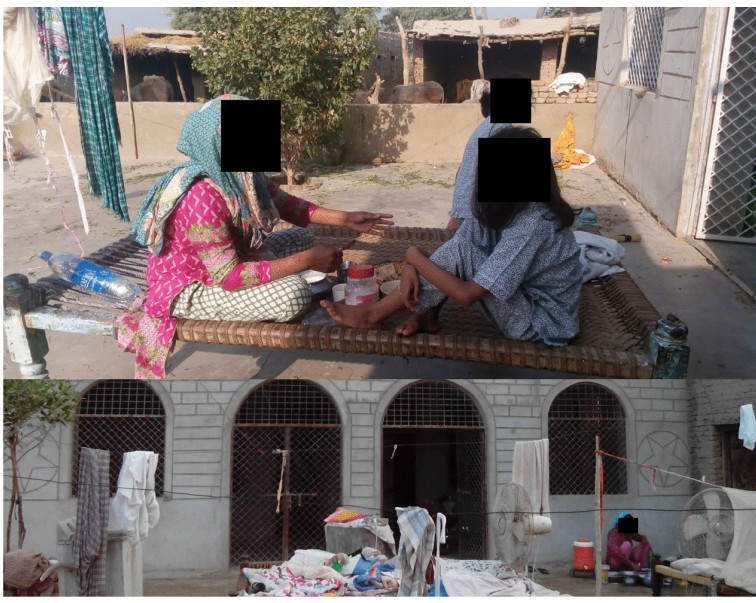

(a)

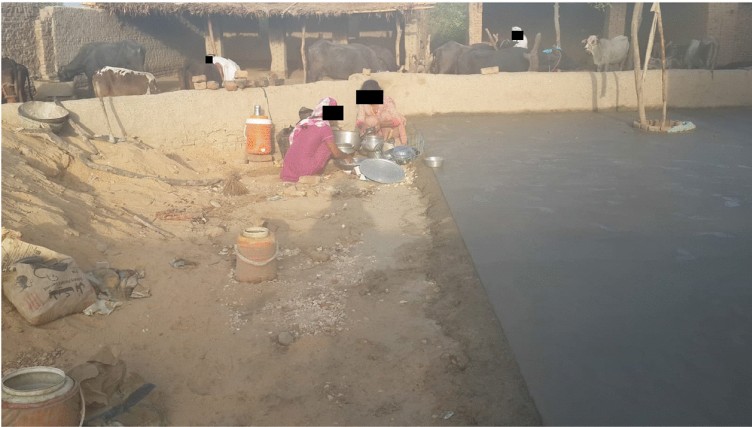

(b)

**Fig 1. A participant's home in a rural area of Pakistan.** The image was obtained during the research process with the permission of the individuals appearing in the photo and is not sourced from third-party copyrighted material. **(a)** Dining and bedroom area of the house. Traditional bed shown in the image used for sleeping and having meals. **(b)** Kitchen and storage area of the house.

## COVID-19 impact and communication technologies

During the COVID-19 pandemic, All European participants reported a decline in QoL due to increased social isolation impacting their mental health. In contrast, participants from rural Pakistan experienced minimal change, due to living with extended family in a small village. All participants except those from rural Pakistan, reported that communication technologies like WhatsApp, Skype, Zoom, and Facetime helped mitigate isolation during the pandemic. Eight Participants (36.36%) working in the IT field or at universities reported having prior experience with these technologies even before the pandemic, making it easier for them to adapt and integrate them into their daily routines. These participants used a wider range of social and communication platforms, such as Slack, MS Teams, Zoom, Facebook, Instagram, WhatsApp, and Skype, compared to participants with non-technical backgrounds. One participant also reported the benefits of engaging

with their doctor over remote communication during COVID. Other studies also reported similar increases in technology use among older adults during the pandemic [25,33]. Post-COVID research indicates that repeated engagement with digital technologies enhanced users' computer self-efficacy, mastery experiences, and confidence, leading to more positive perceptions of ease of use and usefulness [34]. Our findings reflect these patterns too, supporting the view that these determinants, well-established in technology acceptance research may be relevant for understanding older adults' acceptance of smart-home technologies in the post-COVID context.

These findings suggest that technology literacy shaped the extent to which communication tools buffered isolation, but cultural living arrangements (nuclear households in Europe vs. extended families in Pakistan) also strongly influenced outcomes. This again reflects Maslow's hierarchy [35]: while European participants emphasised unmet social belonging needs, Pakistani participants highlighted satisfaction of safety and physiological needs through family proximity.

### Awareness and knowledge about smart technologies

Twelve participants (57%) were unaware of smart home technologies and their benefits. Very few (n = 2) were able to identify features like smart lights and automation tasks (see Fig 2). It was striking to see that many participants were surprised to learn that these technologies are already available and not merely 'futuristic'. All participants from rural Pakistan did not know most smart home components, one was confused as it was something they had never encountered before. Upon discovery, many expressed interest in exploring these technologies.

Six participants (27%) expressed that their "elderly" relatives at high risk of falls, living alone, or with dementia, could benefit from such technologies, if aware. One recalled a relative with dementia who would have benefited from smart glass or smart home technology as they died from dehydration, despite nursing home care. Another stated that their relative refused to use wearable devices (PERS) despite family pressure. The participant (PI500) stated "I think that's very good... you know we had all these problems with her. it would have been so helpful to know certain things … the time she was to fall or forgot her cooker"; Highlighting the concerns and frustrations experienced by relatives. Thus, lack of awareness and availability creates a major hurdle in incorporating these technologies and benefiting older people.

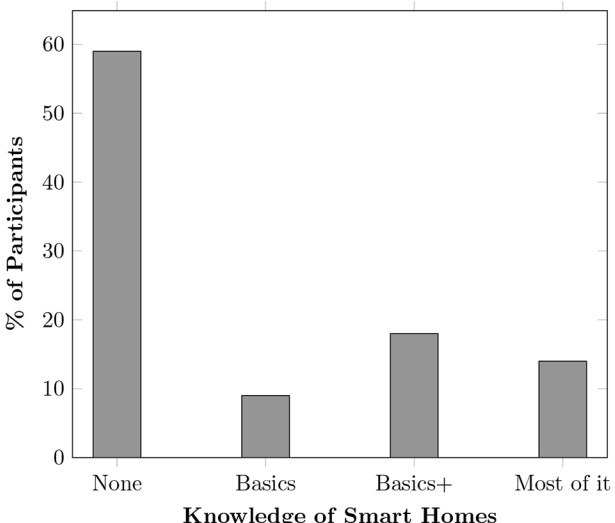

**Fig 2. Participants' knowledge about Smart Homes.** The terms "Basics," "Basics+," and "Most of it" reflect increasing levels of understanding of these technologies, from fundamental operations to advanced system integration and customised smart home systems.

Those with prior exposure to technology, especially in Europe, were more open to adoption, influenced by positive experiences. These participants, having attended public research events such as exhibitions or the iRobot showcase at V&A, or having conducted personal research were able to define smart homes in greater detail and had a positive attitude towards them. Positive attitude was also common among participants for known technology such as PERS which is widely used in the UK and issued by healthcare providers and perceived as trustworthy, reliable, and affordable among UK participants. Participants whose family members actively used smart technologies had a positive opinion from observing the benefits first-hand. However, if initial engagement was negative, older adults were reluctant to continue use. This highlights that engaging older adults with technology can enhance their awareness of its benefits, functionality, costs, and how it can be integrated into their existing homes.

Fig 2 illustrates participants' knowledge of smart home technologies, categorised into four levels, "None," "Basics," "Basics+ ," and "Most of it." Where "None" indicates no knowledge or experience to "Most of it" which represents advanced expertise. Basics refers to a general awareness of common devices like smart bulbs and voice assistants but little hands-on experience. Basics+ represents moderate familiarity, where individuals have used or installed smart devices and understand basic automation. Most of it reflects advanced knowledge, including system integration, customisation, and an understanding of various smart home features that enhance daily life, for example, Geeny in Table 2. This classification helps assess awareness levels and the need for further education. The data reveals that a significant majority of participants (60%) reported having no knowledge of smart home systems. In contrast, only 15% indicated having most of the knowledge. These findings indicate a general lack of awareness and expertise in smart home technologies among the older adult population in this dataset. This suggests a potential need for educational initiatives or more user-friendly smart home systems to facilitate broader adoption and effective use.

Smart plugs and "Geeny Smart Home" were the most popular among participants. However, there were concerns around Geeny, such as the impact on NHS resources and costs associated with private healthcare for medical checkups. Figs 3–5, highlight varying perceptions and adoption trends across different smart home technologies. Fig 3(a) shows smartphones were found to be the most common technology owned among participants whereas smart homes were seen as a valuable future investment. One reason for participants' reluctance to invest in smart homes was their unfamiliarity with the system, its benefits, and concerns about reliability.

Awareness of smart technologies was generally higher in Europe, partly due to exposure at public exhibitions and showcases. Pakistani participants showed more curiosity once informed, but also linked adoption closely with affordability. This pattern is consistent with the Technology Acceptance Model (TAM) [36], where perceived usefulness and perceived affordability both predict attitudes toward adoption. According to this model, people decide whether to adopt a technology based on two main perceptions. The first is how useful they believe the technology will be, and the second is how easy they think it will be to use. These perceptions shape their overall attitude and intention to adopt [4]. This perspective supports the interpretation of the patterns identified in this theme.

### Environment impact

Three participants (14%) expressed a strong preference for technology that was environmentally friendly, energy-saving and reduced carbon footprint. For example, the smart HVAC system was considered to conserve energy, save cost and thereby not exacerbate global warming. A participant even inquired about the manufacturing processes, materials used, energy consumption, and environmental impact before adopting smart home technologies. For example (PI2908) "the technology required ... does that have an impact on scarce resources of rare metals? … worldwide resources are limited".

The issue of obsolescence also emerged. A participant preferred using devices for extended periods, resisting the trend of frequently upgrading to the latest models. This approach stemmed from a desire for sustainability and a belief in repairing and maintaining rather than discarding and replacing. The planned obsolescence of devices particularly was found to be "bizarre" by a participant using the example of Apple devices. European participants associated sustainability with

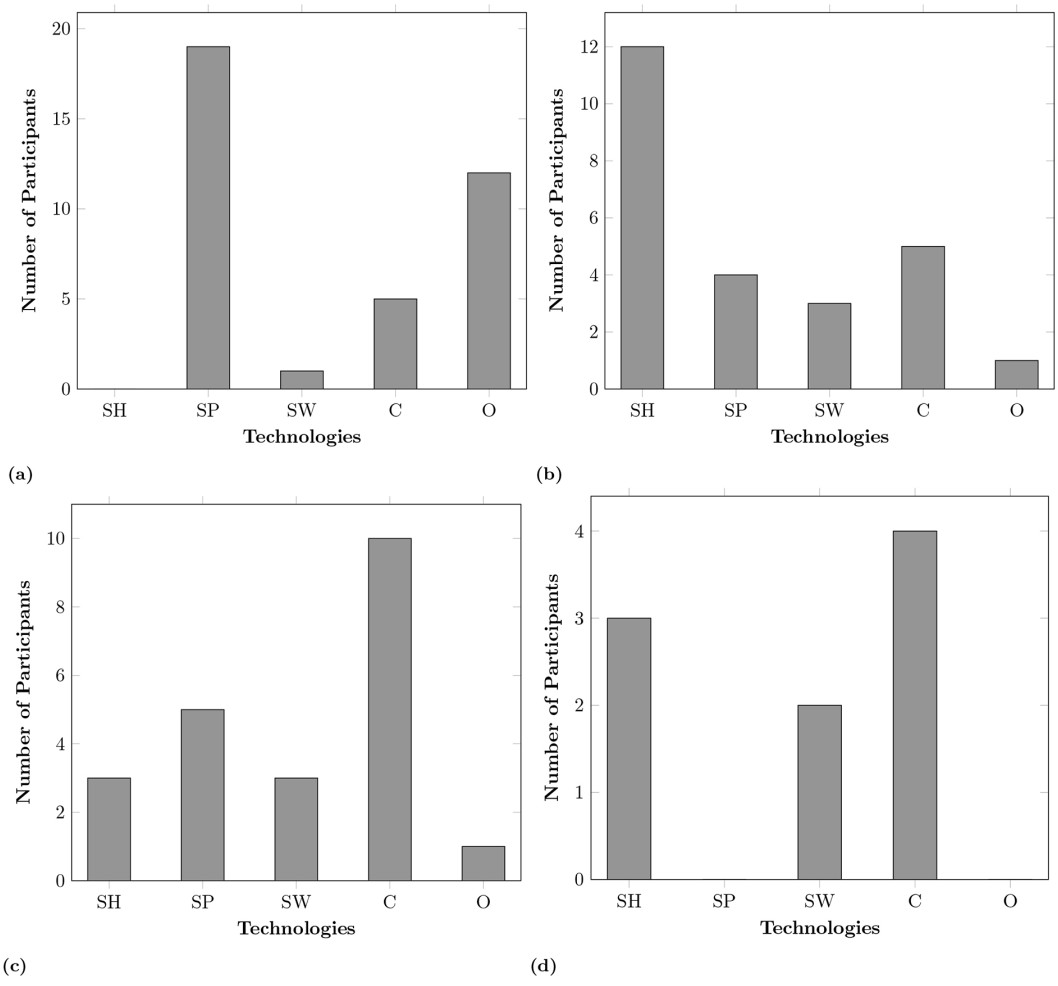

**Fig 3. Technology-related questions, where SH: Fully-integrated smart home, SP: Smartphone, SW: Smart wearables, C: Cameras (security only), O: Other (e.g., smart plugs, lights, meters, assistants, clinical devices, etc.). (a)** Technology owned, **(b)** Willingness to adopt technology, **(c)** Technology perceived to have privacy concerns, **(d)** Technology perceived to provide less control.

energy use and carbon footprint, whereas Pakistani participants did not raise sustainability as a concern. Instead, they emphasised having the possibility to adopt technology in ways that could enhance their QoL.

## Necessity and intended benefits

It is imperative that technology should provide its intended benefit to its users. This was evident in the study where participants expressed varied experiences and perspectives. For example, one participant's negative experience with a faulty smartwatch highlighted the need for basic functionality to tell the time in addition to reminders, recording steps, and mileage; The participant (PI9127) stated "I've just replaced it because the one thing it couldn't do is tell me was time."

Ten participants (52%) were hesitant to adopt smart home technologies as shown in S2 Table in S1 Appendix, often preferring familiar methods, and expressed scepticism about the need for additional smart home devices. They found smartphones sufficient for everyday tasks including reminders, steps etc and that traditional medical devices could be used to monitor health decline. Whereas, other respondents saw the benefits of automated systems such as smart

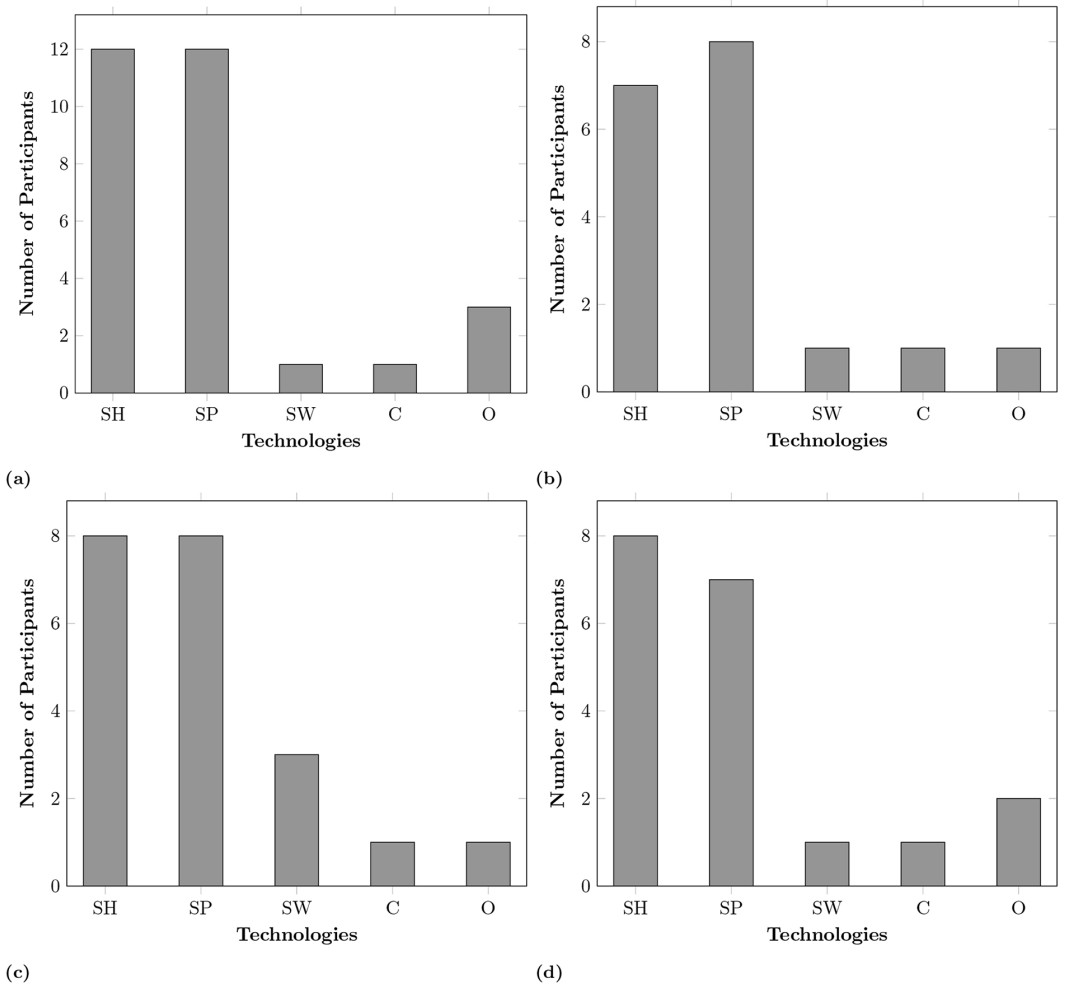

**Fig 4. Technology-related questions, where SH: Fully-integrated smart home, SP: Smartphone, SW: Smart wearables, C: Cameras (security only), O: Other (e.g., smart plugs, lights, meters, assistants, clinical devices, etc.). (a)** Which provides most beneficence for you? **(b)** Which is most unobtrusive? **(c)** Which promotes independence? **(d)** Which promotes autonomy?.

pillboxes, particularly for those with memory impairments. Some participants found that continuous monitoring of everyday life, such as a fridge, was seen as unnecessary. Regarding Mixed Reality, participants perceived the bulkiness and VR's unclear benefits of social connectivity as factors making it less appealing compared to traditional video calls.

It was a recurring theme of the analysis that participants preferred adopting or investing in smart technologies that would be of marked benefit to their lives. For example, a participant (PI2908) stated "I think it's asking somebody my age and I understand why you're doing that, but we haven't been used to that technology for all of our lives and and everything you know, as I said, enjoy a good QoL. There's a little bit of resistance in the sense of, well. Why do I need it?." The multi-functionality of smartphones made them uncertain as to why additional sensors and devices would be needed. Participants stated they would adopt or invest in smart technologies that would be of marked benefit to their lives. For example, one of the participants (PI325) said: "Well, I need to really see what the benefit was because I think, like many people of my age, I'm used to things as they are. I would prefer technology to use it for the problems that I have in life., and I would like it to use it to replace. If I'm happy with things as they are, then I don't see that I would want to replace it."

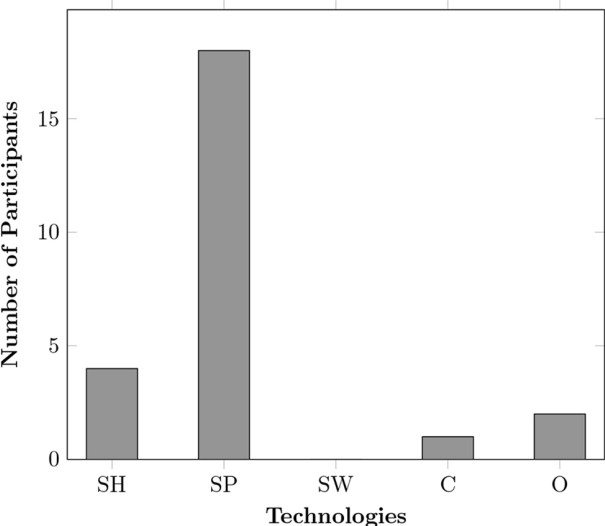

**Fig 5. Technology-related questions, where SH: Fully-integrated smart home, SP: Smartphone, SW: Smart wearables, C: Cameras (security only), O: Other (e.g., smart plugs, lights, meters, assistants, clinical devices, etc). a)** Which supports socialising?.

Compared to people living in Europe, people from Pakistan, especially those from rural areas, were very keen to adopt technologies to monitor and improve their health. One rural respondent (PI725) expressed that "if a person can benefit from these technologies, why would I not want them?... The only question is about their availability and affordability."

Twelve participants (57%) reported being unaware of the first-hand benefits that smart home technologies could offer. European participants who were unaware of these benefits were convinced that their current systems worked sufficiently well. Others acknowledged the potential need for such technologies but remained uncertain about their usefulness. Overall, participants did not appear to perceive clear benefits, highlighting the importance of raising awareness and providing education on how smart homes could enhance daily life. In contrast, participants from Pakistan viewed such technologies as highly necessary and believed they could significantly improve their lives, particularly in rural areas, expressing a strong willingness to adopt them immediately. When participants were shown the "Genny" use case video, they perceived it as significantly beneficial. While acknowledging its potential advantages, European participants emphasised the need for personalisation to align with individual lifestyles and to ensure that social aspects were addressed, as reflected in S2 Table in S1 Appendix, where smart homes were perceived as the least socially oriented. Compared to abstract descriptions, presenting real-life use cases such as the Genny video and enabling users to actively engage with smart technologies proved more effective in fostering positive attitudes and raising awareness of their benefits. Resistance to adoption may decrease when intended benefits and added value are made explicit, and when users feel sufficiently informed to make decisions that could improve their QoL. These findings align with the Technology Acceptance Model (TAM) [36], which suggests that adoption depends not only on perceived usefulness but also on awareness and trust in the technology.

### Accuracy and reliability

Accuracy, reliability and trust in technology emerged as a common concern among participants when considering the adoption of smart home technologies. Overall, 13 out of 21 participants (62%) explicitly raised doubts about reliability or accuracy. Those who had a negative experience put less trust in smart devices and questioned their accuracy, as was the case for the individual mentioned above with the faulty smartwatch.

Participants with reliability concerns reported being more comfortable with simpler, more reliable solutions than smart technologies. Specifically European participants preferred traditional technologies such as pressure pads used by housing associations for older adults or for example, PERS which was perceived as reliable despite not being physically appealing and expressing stigma-related concerns.

Similarly, participants shared experiences of smart technologies including smartphones failing to perform as expected, making them cautious about investing in smart devices. Reliability concerns related to battery life and biometric technology were common, with one participant citing inconsistent fingerprint recognition on their iPad. Due to these reasons, they expressed reservations about the dependability of smart doors and pillboxes. Accuracy was another significant concern, especially in technologies for health and location monitoring. For example, a participant shared an incident where a phone's tracking feature provided inaccurate location data, causing distress to find a relative, and leading to a negative perception. Similarly, concerns were raised about the potential for human or machine error in more complex systems like smart homes, with participants recommending extensive testing and validation of smart technologies before being released. Participants were willing to excuse the look and feel, provided the smart technology was accurate, and reliable and delivered the intended benefits.

European participants and urban Pakistani participants stressed that smart technologies must be accurate and reliable to be adopted. In contrast, rural Pakistani participants did not explicitly raise issues of reliability or accuracy, instead taking taking a straightforward view stating: "why wouldn't I want it if it helps me?" which reflects a stronger focus on perceived usefulness than on reliability or accuracy, possibly due to their limited first-hand experience with such technologies.

### Anxiety and fear

Anxiety emerged as a significant barrier to technology adoption, with some participants worried about over-dependence or becoming overly anxious about their health, particularly in decline. Only Eight European participants (38%) described feelings of stress or worry linked to continuous monitoring or reliance on devices. Financial anxiety also deterred users, for example in the case of a participant with expensive hearing aids, where the fear of loss or damage, exacerbated by daily activities increases the risk of damage, leading to limited use. Conversely, another user found digital hearing aids life-changing, praising their ease, unobtrusiveness, and phone integration, which significantly improved their QoL. These contrasting experiences with hearing aids illustrate how technology can either enhance life or cause anxiety, affecting its adoption and integration into daily routines. European participants were more likely to describe emotional anxiety linked to health monitoring, whereas Pakistani participants described practical or financial anxieties. This distinction reflects the influence of socioeconomic context on how technology-related fears are framed.

### Losing human connection

Concerns about smart home solutions reducing personal contact were prevalent, with fears that technology reliance might diminish interactions with loved ones, despite its use for family reassurance. Only three participants (16%) felt that smart homes could support socialising, while the majority perceived them as isolating and raised concerns about reduced social interaction. Participants in the study identified themself as very sociable and felt strongly about building trust and having social communities. They felt the COVID-19 pandemic intensified the need for physical presence, underscoring technology's limitations in replacing human interaction. A participant (PI4278) stated: "It's so long as it doesn't become an excuse for not having personal visits".

By contrast, Pakistani participants did not generally express this concern, likely because cultural and family structures already ensured frequent daily contact.

One participant in the UK had creatively used technology to maintain safety and connection within their social circles. They described a system where friends send daily messages to check on each other. If someone doesn't respond, a member of the group visits to ensure their well-being. This strategy was used to help a friend in time who had a fall and

was not able to communicate it. This was common among other participants who communicated to their family members and friends via smartphones in emergencies. It is crucial to design smart home solutions that can monitor health and safety within a home but also promote community engagement by integrating family and friends into the monitoring system, preventing isolation. Overall, European participants worried about losing human connection, while Pakistani participants viewed technology as complementary to, rather than replacing, family support. This reflects cultural differences in how technology is situated within social relationships.

**Affordability**

Cost was a significant factor in not adopting smart home technologies [37,38]. While some UK participants were willing to spend as much as needed to stay independent rather than enter expensive residential care (Fig 6). Others felt that most of the technology was costly including updating and replacing it to keep up with evolving technology. Additionally, the expense of integrating smart objects into older buildings was found to be challenging, highlighting the need for systems that can be deployed affordably and with minimal structural alterations regardless of a building's type and age. Participants from a low-income background in Pakistan had more concerns about cost than European respondents. All rural Pakistani and one urban participant identified affordability as the primary barrier, in contrast to seven European participants. For them, the affordability of smart technology was critical, with a willingness to invest only in reliable and long-lasting solutions within a specific budget (up to £172 at the time of the interview). The highest value for a European outlier was up to £75,000. Figs 6 and 7 show the affordability box plot over all participants and country-wise. It is quite evident that people from rural areas of Pakistan have a much smaller range compared to that of urban areas and Europe. However, most participants were willing to spend more on smart homes, smartphones and then cameras compared to the rest of the smart home technologies. Those without prior knowledge of technology costs were open to spending if the need arose, unlike low-income individuals; for example, a participant (PI894) stated: "I cannot afford this at its original price, however, if these smart devices come to the local 'Bazaar', once they have been discarded ..from first world countries, we can get them second hand at a more affordable price and still reap their benefits". If the cost of investing or maintaining smart technology is very high, then it would primarily benefit higher-income households, although health is a basic right. It is important that the local governments, healthcare providers and smart home solution providers create cost models and

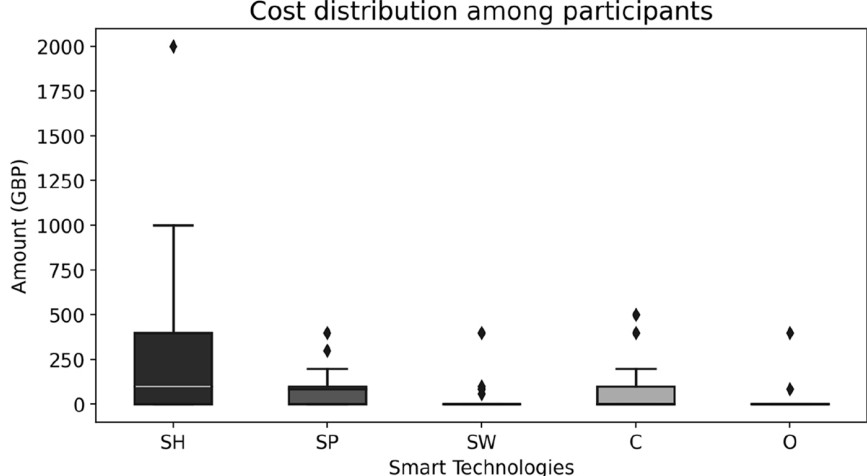

**Fig 6. Participants willingness to spend on smart technology (excluding an outlier of GBP 75,000), where SH→Smart Home, SP→Smart Phone, SW→Smart Clothes/Wearables, C→Cameras (security only), O→Other.**

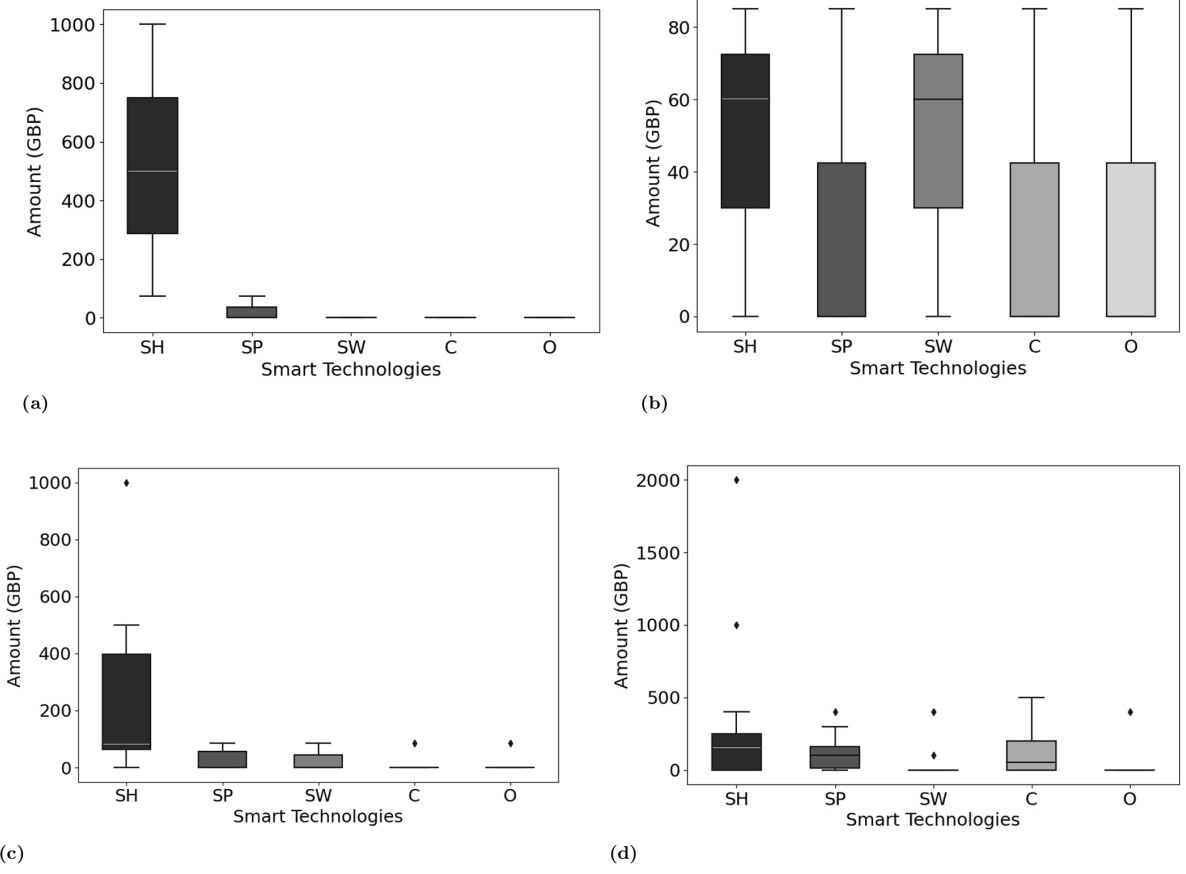

**Fig 7. Participants willingness to spend on smart technology in Pakistan and Europe, where SH: Smart home, SP: Smartphone, SW: Smart wearables, C: Cameras (security only), O: Other. (a)** Urban Pakistani participants affordability, **(b)** Rural Pakistani participants affordability, **(c)** Urban/Rural Pakistani participants affordability, **(d)** European participants affordability.

policies to allow low-income individuals to afford them. These findings highlight that affordability was a universal concern but it was particularly salient among rural Pakistani participants, for whom cost represented the primary barrier to adoption. This suggests that while European participants framed affordability as one of several factors shaping uptake, in rural Pakistan it emerged as the decisive determinant of whether smart technologies could realistically be adopted.

### Design challenges and ease of use

Participants faced challenges with smart technologies like smart meters, smartphones, smartwatches and ECGs particularly due to their complex designs and interfaces, leading to discomfort and inconvenience. A participant (PI4278) stated: "A lot of the technology... is all about miniaturisation. As you get older, your manual dexterity diminishes, and your ability and the logic required to operate the device... my son... who's got a smartwatch. And he was unable to operate it because he hadn't had time to sit down and go through the instruction book to figure out how to work it. And I think that would be an issue." This suggests that technology should be designed for easier implementation, minimising the need for extensive setup or learning. Instead of requiring users to navigate complex instructions, smart home solutions could come with integrated support services or be bundled as complete, ready-to-use packages. Without such considerations, the effort required from users may become overwhelming, diminishing their willingness to adopt the technology.

Participants with IT backgrounds also highlighted the challenges of small screens, poor readability, and confusing User Interfaces (UI) suggesting that non-technical individuals might struggle more. There was a strong preference for user-friendly, configurable technology. Even cost-effective devices were considered ineffective if complicated to use. The importance of visual clarity for those with eyesight issues was emphasised, along with non-user-friendly concerns over app and website interfaces. One summarised (PI4278) it as "I think it comes back to one's ability to operate phone [...] How big the buttons are, the ease of use, logic attached to the operating system ... things have to be simpler."

Simple, well-designed apps were favoured over those with bugs or poor designs, with some preferring traditional methods, like paper prescriptions, due to digital frustrations. When discussing comfort, a participant (PI200) from Pakistan expressed that "..people in Pakistan sadly face a lot of challenges on a daily basis, but this makes them resilient as discomfort is unavoidable." Hence, any technology that will improve aspects of life would be beneficial and accepted, despite being uncomfortable.

Overall, 7 out of 14 European participants mentioned ease of use and design challenges. While rural Pakistani participants did not discuss usability directly, likely due to limited exposure to such technologies. Among urban Pakistanis, a participant described a higher tolerance for discomfort as long as technology brought tangible improvements. This contrast suggests that European participants prioritised design and ease of use, whereas Pakistani participants focused more on the availability of technologies and the tangible benefits they could provide, reflecting cultural and socioeconomic resilience. This aligns with Diffusion of Innovations [39] and Maslow's hierarchy of needs [30], suggesting that in resource constrained contexts adoption is driven less by usability and more by accessibility and the ability to meet basic health and safety needs, whereas in more resourced settings the focus shifts toward comfort, convenience, and autonomy.

## System integration, compatibility and set-up issues

Three participants (14%) configured, bought and set up smart devices themselves without additional help. In addition, seventeen other participants (81%) described independently configuring non-smart technologies such as laptops, phones, or cameras (Table 4). Those who already had smart home components (n = 3 as shown in Table 4) were self-motivated and not swayed by external pressure to adopt the technologies. After their interview, many expressed their willingness to investigate the technologies discussed during the interview and invest when the need arises (n = 10 as in S2 Table in S1 Appendix). However, a challenge noted was the lack of compatibility and integration between different smart devices and operating systems. For instance, a participant who installed smart plugs found each brand had a different setup method, highlighting the lack of standardisation. Even those comfortable with technology, like setting up a new laptop, faced difficulties due to these inconsistencies. This varied process for each new device complicates the user experience.

Compatibility concerns also extended to physical installation. such as smart doors, and HVAC, for installation within older buildings which can be complicated and costly but also disrupt the aesthetic of older houses. Cultural contrasts emerged where European participants primarily emphasised interoperability and ease of use, often describing frustration with varied setup processes. In contrast, Pakistani participants specifically those in rural areas highlighted availability and access, noting that these devices/systems were not obtainable.

These findings suggest that adoption is shaped by the quality of the user experience, particularly interoperability, reliability, and ease of installation. Early negative experiences with complex set-ups or poorly integrated systems can discourage users and undermine confidence in smart technologies. Notably, older adults in this study demonstrated that they are capable of configuring and installing devices, the barrier lies less in their ability and more in the unnecessary complexity of installation processes. However, this capability was not observed among rural Pakistani participants, which could be due to lack the technical knowledge or simply a lack of access to these technologies which needs further study.

System designers and engineers should allow the devices to integrate into an existing home rather than asking them to completely replace a previous system, be reliant on a specific company for all the products or go through complex

technical steps to figure out how to use the system. This will only discourage users and create a negative experience resulting in low adoption.

### Over-monitoring

Five participants thought that smart home solutions were over-monitoring every detail of life, which was not necessary. Participants used terms like "not necessary," "neurotic," and "hypochondriac" to define over-monitoring behaviour. Participants expressed concerns related to over-monitoring such as developing an over-dependence on health monitoring where one questioned the balance between need and want, especially regarding camera monitoring. Participants strongly felt about consent for monitoring and allowing them to have control over their lives when using these systems. In contrast, Pakistani participants did not view continuous monitoring negatively with the exception of two urban participants who expressed concerns regarding security and data privacy. They accepted it as part of modern life and valued the possibility of receiving immediate help when needed. This cultural difference may suggest that while European participants associated monitoring with concerns about autonomy and privacy, Pakistani participants linked it to the need for urgent assistance, particularly in rural areas where access to healthcare is already limited. These findings highlight how cultural values shape perceptions of technology, aligning with cross-cultural technology adoption theories such as TAM [36], where adoption is influenced not only by perceived usefulness but also by contextual factors like trust, accessibility, and cultural norms.

### Health monitoring in an emergency

Participants living independently (alone or with their partner) thought smart homes were an improved way of monitoring the health of individuals at home in an unobtrusive manner; given the system is validated, tested for its reliability, caters to ethical concerns and is of reasonable cost, it would be the most suitable solution and provide peace of mind to their children and other family members. They found value in systems like Geeny for immediate emergency alerts, benefiting those with nearby families. However, participants within the UK living alone questioned the monitoring responsibility. Another significant issue was the effectiveness of alerted health services, especially in remote areas where ambulance delays compromise system reliability. For example, a participant (PI2090) said: "It's the speed of response by the ambulance service, which might be the limiting factor in a perfect system." This was a common problem found among many participants where they expressed that during an emergency situation such as a fall faced by them or their loved one. Participants mentioned that they somehow managed to drive (with a ruptured tendon) to their partner for help, call using a smartphone or, in most cases, (fractures or other issues) wait for their family member to arrive home. These experiences of self-managing emergencies highlighted the limitations of existing emergency infrastructures. This suggests that smart systems should not be viewed in isolation but rather as part of a broader integrated infrastructure. Effective implementation requires considering various factors, including regional differences, accessibility, and the ability to connect with existing emergency response systems. Participants from the UK residing in remote areas often highlighted the limitations of public services, such as delayed ambulance responses, while rural Pakistani participants described difficulties in accessing healthcare services. In emergencies, participants in both settings reported relying primarily on family support. The findings indicate that limited access to emergency services poses the greatest risk in contexts where such services are most urgently needed, highlighting the importance of technologies that can help address these gaps.

### Privacy and other ethical concerns

The most prominent ethical concerns and the widely researched themes are privacy and data security [4], which participants also strongly emphasised in this study. The results from our study also show that privacy was the primary concern among participants, as shown in Fig 8. All European and urban Pakistani participants expressed strong privacy concerns, while rural Pakistani participants were less concerned, perhaps reflecting limited exposure to these technologies (Table 4).

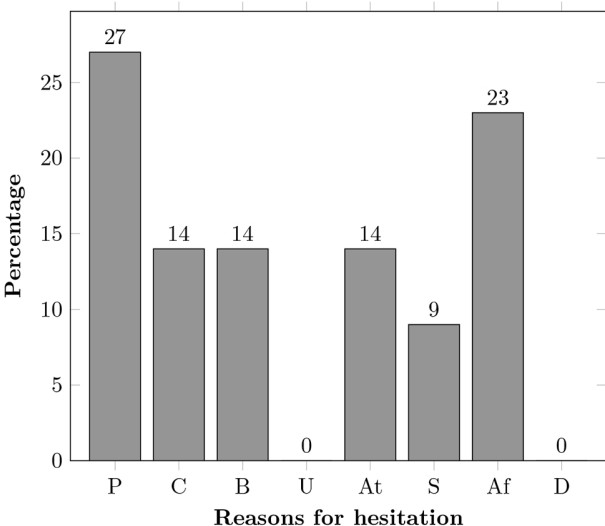

**Fig 8.** Reasons for hesitation in adoption of technology among all participants arranged by the first rank order only (where P: Privacy, C: Control, B: Beneficence, U: Unobtrusiveness, At: Autonomy, S: Supporting Supporting socialisation, Af: Affordability, D: Discomfort).

Participants were generally reluctant to install indoor cameras, preferring PERS for family health monitoring. The decision to allow camera monitoring was seen as dependent on the relationship with the potential observer. While some were open to outdoor security cameras, most were opposed to internal monitoring, valuing their privacy regardless of age.

Ethical considerations extended beyond personal privacy. Participants were apprehensive about sharing additional personal data, highlighting concerns regarding privacy and data security including the need to communicate related policies in plain language. They emphasised the importance of knowing where and how securely data is stored. Participants valued their independence and control over privacy and data sharing. They wanted the ability to configure data-sharing options and have full control over the system. They did not feel every detail about their health or activity needed to be shared. Some participants expressed reluctance to purchase products from companies with unethical practices, showing a preference for brands that aligned with their ethical views and prioritised environmental friendliness and data privacy.

Participants showed a preference for ethical brands, valuing environmental sustainability, and data privacy, and avoided companies with unethical practices. Only a few participants had some of the smart home components installed within their homes which they thought did not hinder their privacy.

Overall, privacy concerns were dominant among Europeans and less pronounced among Rural Pakistanis, suggesting that cultural orientation and socioeconomic context influence which risks are prioritised in smart home adoption. This difference aligns with broader cultural frameworks, where individualistic societies emphasise autonomy and personal control [40], while in our study collectivist societies showed greater acceptance of shared oversight.

## Discussion

Smart technologies enable older adults to benefit from various features to improve their QoL and live independently within the comforts of their homes. From health monitoring to security and convenience, several smart home products are available in the market and in research that can fulfil the needs of older adults. However, despite this growing availability, our study found that most participants were unaware of these technologies or their potential benefits for themselves and their loved ones. This lack of awareness presents a significant barrier to adoption as people are unlikely to invest in or engage with technologies they do not fully understand or recognise as valuable. As shown in Table 4, while many older

adults use smartphones, laptops, and communication technologies, the adoption of smart home components remains significantly lower, particularly outside of Europe. Those with some awareness as shown in Fig 2 mostly recognised basic smart devices, such as smart bulbs and voice assistants, rather than the more integrated and complete solution systems designed to enhance daily living. This highlights a critical gap in awareness, suggesting that educational, training efforts [41] and more intuitive designs could play a key role in making these technologies more accessible. This finding resonates with TAM [36], where perceived usefulness and perceived ease of use are central to adoption. Older adults' limited awareness undermines these perceptions, reducing the likelihood of adoption.

This study found several reasons behind this reluctance as shown in Fig 8 and S3 Table in S1 Appendix. Privacy concerns stood out as the primary hesitation factor, with 27% of participants reluctant about sharing their data and physical privacy. Affordability was another major concern, with 23% of participants feeling that investing in new technologies was too costly, particularly given the frequent need for upgrades. This challenge was even greater for participants from underprivileged backgrounds or lower-income countries, where limited financial and infrastructure support restricted access to smart home technologies. As shown in Fig 1, inadequate structures and resource constraints made adoption difficult. An interesting finding was that people from rural areas were more inclined to adopt technologies that could assist in emergencies or improve their QoL, even if it meant compromising privacy. This may be due to limited awareness of data privacy or a greater need for health improvements in underserved areas. These dynamics reflect constructs of UTAUT2 [42], where cost and facilitating conditions strongly shape adoption. UTAUT2 explains technology adoption through several key perceptions, including performance expectancy, effort expectancy, social influence, facilitating conditions, hedonic motivation, price value and habit. In this context, cost and the availability of supportive conditions appear influential, which aligns with the model's emphasis on perceived value and practical enablers [4]. Moreover, the rural participants' willingness to compromise on privacy illustrates a contextual trade-off between perceived risks and perceived benefits, echoing sociotechnical perspectives. Comparable studies have shown that privacy and affordability concerns frequently dominate older adults' decisions about smart home adoption which aligns with our findings [4,43,44].

Despite these concerns, the study also revealed positive attitudes toward certain technologies. Smartphones, for instance, were seen as crucial for socialising, with 72% of participants viewing them as a key tool for staying connected with others. Smart homes, while less commonly used, were recognised for their potential to support independence and autonomy, making them a worthwhile investment for the future. Wearables and security cameras on the other hand received lower adoption and trust, likely due to concerns about privacy, comfort, stigma and ease of use. The study found that even though older adults are hesitant now, they recognise the long-term value of these systems if key barriers are addressed. This is consistent with the Diffusion of Innovations theory [39], in which technologies perceived as compatible with existing practices spread more easily than those perceived as intrusive or stigmatising.

This also aligns with our literature review, which found a strong preference for systems that offer user control over fully automated solutions [4]. The ability to manage data and system settings provides a sense of security and personalisation, reinforcing trust in smart home technologies. Similarly, research consistently highlights the importance of adaptability and customisability where older adults prefer systems they can adjust to their needs, and they tend to reject rigid or fully automated solutions that undermine autonomy [4,45]. Participants also showed they overcame their difficulties in connecting with families using communication technologies, highlighting the increased reliance on technology during the pandemic [46]. This demonstrates that when the value of technology is clear and immediate, adoption rates increase. From a theoretical perspective, this underscores the importance of perceived control, which extends TAM [36] and Elderadopt [47] by emphasising autonomy as central to older adults' adoption decisions.

One unexpected finding was how marketing and messaging around smart home technologies can discourage adoption. Many older adults do not perceive themselves as frail or vulnerable, yet these technologies are often marketed as solutions for the elderly. Our research found that when products were associated with weakness or dependency, participants were less inclined to adopt them [27]. This suggests that rethinking the way smart technologies are presented,

emphasising empowerment, convenience, and lifestyle enhancement rather than ageing and vulnerability could encourage broader acceptance. This extends literature on framing effects and TRA/TPB [48,49] showing how identity and subjective norms influence behavioural intention alongside functional evaluations.

Our findings suggest that older adults prefer smart systems that offer user-friendly interfaces, intuitive controls, and integration into daily routines. Technologies should be designed to minimise disruptions rather than introduce complexity. Participants emphasised that smart home solutions should provide tangible benefits to their health and well-being, rather than merely adding unnecessary features. Familiarity with technology also played a role in openness to adoption. Those already comfortable with smartphones or personal alarm technologies (PAT) in the UK, for instance, were more willing to explore smart home solutions due to their positive experiences with existing devices. Maintaining independence was a primary motivator, with participants open to adopting smart technologies if health declined or due to family concerns [50]. However, they prioritise privacy and social connection to enhance, rather than replace, human interactions. They preferred smart homes that facilitated engagement with their communities rather than promoting isolation [4,50]. This suggests that designers should focus on integrating social connectivity features within smart home systems to align with older adults' values and preferences. This finding also resonates with the Capability Approach, where technologies are valued not just for functionality but for expanding real freedoms and opportunities for well-being [51].

Our study found an interesting insight that culture may play a role in the adoption of technology. In many Asian communities, it is common for families to live together in either unitary or multigenerational households, where family influence on older adults' technology adoption can be seen as positive [52]. Intergenerational relationships in Asian countries may encourage the use of smart home services and the adoption of new technologies [53]. Technology adoption may also increase if the older adult has tech-savvy family members, possibly because of the exposure, support and guidance [54]. These dynamics align with TAM2 [55] and TPB [49], where social influence and subjective norms shape technology acceptance. They can also be framed through cultural models such as individualism-collectivism in collectivist contexts where family interdependence promotes shared decisions, while in more individualist contexts adoption is seen as a matter of autonomy [56,57]. By systematically comparing these cultural logics, our study shows how family-based collectivist norms in Pakistan foster adoption through shared responsibility, while individualist orientations in Europe emphasise autonomy and self-reliance, sometimes leading to slower uptake. This structuring of cultural influence represents a key conceptual contribution beyond descriptive accounts. It is important to note that the largest proportion of participants in our study (47.6%) were aged between 65 and 70 years (Table 3). Representation decreased in the older age brackets, with 28.6% aged 71–75, 19.0% aged 76–80, and only one participant (4.8%) in the 81–90 age range. Geographically, the majority were based in the United Kingdom (61.9%), with smaller groups recruited from Pakistan (urban: 14.3%; rural: 19.0%) and Malta (4.8%). Gender distribution was slightly skewed, with males comprising 57.1% of the sample compared to 42.9% females. These demographic characteristics highlight both the strengths and limitations of the study. While the sample provided valuable insights into the perspectives of older adults, its size and geographic concentration limit the generalisability of the findings. As our sample was skewed toward UK participants, and while this provided rich insights into western contexts compared to Pakistani context including it rural areas, the findings may not fully capture the diversity of cultural influences globally. Future research would therefore benefit from including larger and more diverse cohorts, especially from under-represented cultural settings, to capture a broader range of attitudes and priorities for better adoption of smart home technologies.

Participants were willing to adopt technologies when they offered clear practical benefits and aligned with their existing routines and sense of identity, which corresponds well with TAM, UTAUT2, and Diffusion of Innovations. At the same time, concerns about privacy, cost, and the availability of support closely reflected established constructs such as perceived risk, price value, and facilitating conditions [4]. Cultural expectations and family involvement also shaped decisions in ways that align with broader ideas about individualism, collectivism, and shared decision-making. Overall, these frameworks provide a coherent explanation for the adoption patterns we observed and complement the more detailed theoretical discussion presented earlier.

To support older adults in adopting smart home technologies, a collaborative approach is essential. Engaging stakeholders, older adults, designers, families, and caregivers in focus groups and co-design sessions helps tailor technology to real user needs [4,41]. Integrating smart home technology with smartphone applications could improve accessibility, as many older adults are already comfortable using mobile devices. Beyond design, accessibility and affordability are vital. Additionally, showing real-life use cases and allowing older adults to engage with various smart technologies may foster a positive attitude and improve adoption rates. Resistance to new technology tends to decrease when users clearly understand its intended benefits and have enough knowledge to make informed decisions about improving their QoL. While many systems remain out of reach for those in rural or less affluent areas, government support can make a difference. Programs aligned with NHS, for example, can help offset costs and reduce pressure on healthcare resources. Policies focused on these goals, with cybersecurity protections in place, are necessary to ensure safe and equitable access to technology for all. Encouraging adoption involves more than just making technology available; it's about supporting user needs and enhancing QoL which can help bridge the gap between potential and practical use. Additionally, clear policies on data ownership, supported training, and partnerships with organisations make this vision achievable, creating a technology landscape where all can feel secure, supported, and empowered to live with independence.

## Limitations

The research has limitations including a small sample size of 21 older adults, primarily from the UK, which restricts the broader applicability of the findings. Due to the COVID-19 pandemic, the research had to be conducted online, which meant that participation was limited to those who had internet access and were comfortable using digital tools. This excluded older adults with lower digital proficiency, whereas the findings focused more towards older adults with technology familiarity. As a result, the findings primarily reflect older adults already familiar with technology. Additionally, only one participant was over 80, further limiting subgroup analysis. These groups may face greater usability challenges, and future research could address these gaps by using alternative recruitment methods to reach a more diverse and representative sample. Due to the small number of Maltese participants, country-level comparisons could not be done in depth. Therefore, Malta and the UK were combined into a single 'Europe' category. No gender differences were observed in this study. Additionally, age and socio-economic characteristics were not matched across countries, and income related differences were explored qualitatively rather than through stratified comparisons. Larger and more balanced samples across regions and genders would enable more detailed comparative analyses. Given the limited timeframe and reach, this study is an initial exploration, and future research needs to focus on gathering data from a larger, more diverse sample that includes individuals with varying levels of technological experience, from different geographic regions.

Another limitation concerns the availability of the smart home devices discussed. Most of these technologies are manufactured in the EU or USA and are primarily marketed to Western consumers. At the time of this study, such devices were not readily available in Pakistan, particularly in rural areas, which limited participants' direct familiarity or hands-on experience. This restricted exposure may have influenced their ability to fully assess the impact of these devices on quality of life. Consequently, their feedback reflects perceptions and assumptions rather than extensive firsthand use, which should be considered when interpreting the findings. Future studies could address this limitation by including older adults' charities and networks and providing participants with demonstrations or trial access to the technologies, allowing for a more grounded evaluation.

## Conclusion

In conclusion, this study highlights both the promise and the challenges of adopting smart home technologies among older adults. The findings indicate that while these technologies have the potential to improve QoL by enhancing health monitoring, safety, and independence, their benefits are not yet fully realised due to limited awareness and persistent concerns.

Other barriers included privacy concerns, affordability, availability, environmental impact, ease of use, social connectivity and access to emergency support.

Our findings suggest that older adults prefer user-friendly, customisable solutions that integrate into their daily routines without compromising their independence. Cultural influences, family involvement, and prior technology experience also play important roles in adoption. Beyond these empirical insights, the study contributes conceptually by illustrating how technology acceptance among older adults is shaped not only by individual-level factors (e.g., usability, cost) but also by socio-cultural contexts and intergenerational dynamics. This extends existing adoption models by emphasising the interplay between personal autonomy and relational support. To bridge the gap between potential and practical use, it is essential to promote awareness, provide training, and design smart technologies that empower users rather than reinforce dependency. A collaborative approach that actively engages designers, caregivers, policymakers, and older adults can drive more inclusive and accessible solutions. Future research should expand to diverse populations to better understand cultural and regional differences in adoption. With thoughtful design and supportive policies, smart home technologies can move beyond mere emergency tools to become valuable enablers of independence, well-being, and community connection for older adults. Finally, by positioning smart home technologies as enablers of autonomy and social participation rather than solely emergency tools, this study underscores their broader role in advancing active and dignified ageing.

## Supporting information

**S1 Appendix. Interview format and collated data.**
(PDF)

## Author contributions

**Conceptualization:** Pireh Pirzada.

**Data curation:** Pireh Pirzada.

**Formal analysis:** Pireh Pirzada.

**Funding acquisition:** Pireh Pirzada, Adriana Wilde, David Harris-Birtill.

**Investigation:** Pireh Pirzada.

**Methodology:** Pireh Pirzada.

**Project administration:** Pireh Pirzada.

**Resources:** Pireh Pirzada.

**Software:** Pireh Pirzada.

**Supervision:** Adriana Wilde, Gayle H. Doherty, David Harris-Birtill.

**Validation:** Pireh Pirzada.

**Visualization:** Pireh Pirzada.

**Writing – original draft:** Pireh Pirzada.

**Writing – review & editing:** Pireh Pirzada, Adriana Wilde, Gayle H. Doherty, David Harris-Birtill.

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
