## [Decision Letter · Decision Letter 0]

29 Aug 2025

Dear Dr. Pirzada,

We look forward to receiving your revised manuscript.

Kind regards,

Dr Buna Bhandari

Academic Editor

PLOS ONE

Journal Requirements:

This was part of PHD Funded project by the University of St Andrews.

4. Please remove all personal information, ensure that the data shared are in accordance with participant consent, and re-upload a fully anonymized data set.

Additional guidance on preparing raw data for publication can be found in our Data Policy (https://journals.plos.org/plosone/s/data-availability#loc-human-research-participant-data-and-other-sensitive-data) and in the following article: http://www.bmj.com/content/340/bmj.c181.long....

Reviewers' comments:

Reviewer's Responses to Questions

**Comments to the Author**

1. Is the manuscript technically sound, and do the data support the conclusions?

Reviewer #1: Yes

Reviewer #2: Yes

2. Has the statistical analysis been performed appropriately and rigorously?

Reviewer #1: Yes

Reviewer #2: Yes

3. Have the authors made all data underlying the findings in their manuscript fully available?

Reviewer #1: Yes

Reviewer #2: Yes

4. Is the manuscript presented in an intelligible fashion and written in standard English?

Reviewer #1: Yes

Reviewer #2: Yes

Reviewer #1: I hope this letter finds you well. I had the opportunity to review your article titled, “Understanding older adult’s perception, acceptance, and adoption of smart home technologies”, which was submitted the Plos One.

1. Abstract

- The abstract clearly suggests the need for research on the adoption of smart home technology by the elderly.

- In particular, the fact that it clearly raised awareness of the problem in connection with changes in quality of life (QoL) after COVID-19 is a timely and academically and socially persuasive starting point.

- It briefly explains that this is a qualitative study based on online interviews with 21 older adults, and briefly mentions the age range of the participants, region, and interview method.

- Recommendations for smart home technology designers and researchers (e.g., affordability, environmental impact, cultural differences, etc.) are valid and have practical implications.

- However, the results summary remains in rather general sentences and lacks quantitative or theme-focused summaries.

- And there is a problem with the suggestion that by compressing too much content into one paragraph, the core message becomes weak.

2. Introduction

- The introduction presents the necessity of smart home technology as a technological solution to improve the quality of life (QoL) of the elderly, starting from the advent of an aging society and the resulting increase in the medical burden.

- The timeliness and importance of this research are highlighted, particularly against the backdrop of social isolation and limited access to healthcare brought about by the COVID-19 pandemic.

- There appears to be an attempt to explain the differences from previous studies by mentioning positive attitudes (e.g., references [9–14]) and factors hindering technology adoption (e.g., privacy, trust, etc.) in existing studies.

- What is unique about the research objectives is that they describe technology adoption in a post-COVID-19 environment, cultural context (multi-country target), and interview structure centered on smart home commercial technologies.

- The sentence structure of the introduction is clear and logically explained.

- However, although the expressions ‘limited research’ or ‘gap’ are repeated, there is a lack of critical consideration of which aspects are not specifically addressed in which literature.

- Although the research design includes a comparative perspective across regions and cultures (e.g., Europe vs. Pakistan), the introduction lacks a comparison with prior research on this.

- There are many sentences that repeat the research objectives, resulting in unnecessary duplication, and it is judged that there is insufficient distinction between the purpose statement and the literature review.

- Although it has professionalism and originality, the sentences are somewhat long and the main message is not clearly expressed.

3. Method

- The adoption of online interviews and ‘thematic analysis’ as qualitative research methods clearly explains the purpose of seeking to deeply understand the elderly’s perception of and acceptance of smart home technology.

- Comparative analysis that considers the cultural diversity of various countries (UK, Malta, Pakistan) is also evaluated as a strength in the design.

- The approach based on the interview language (English, Urdu, Sindhi) is considered positive as it is designed with cultural and linguistic sensitivity in mind.

- The three-stage interview structure, including online interviews via Microsoft Teams, presentation of image and video materials, Q&A, and a final questionnaire, was systematically designed.

- It is judged that the explicit description of the visual materials and interaction methods provided a real sense of immersion.

- However, the sampling method is based on volunteer responses, and there's a high possibility that the study will be biased toward older adults with digital capabilities. While the researcher acknowledged this limitation, a specific response strategy is lacking.

- In qualitative research, it is essential to mention the clarity of the analysis procedure, replicability, and reliability-enhancing strategies (RQDA, inter-rater agreement, etc.).

4. Results

- The results section is organized into 13 main topics, each with clear subheadings for easy readability.

- Each topic includes participant quotes (direct speech), diagrams (Figures 2-8), and comparative explanations, demonstrating an effort to concretize the interpretation of qualitative data.

- “Environmental impact,” “loss of human contact,” “distrust of technology,” “overmonitoring,” and “design issues” appear to have been extracted based on participant narratives and are compelling, realistic themes based on real-world experiences.

- Figures 2-8 supplement the text by visually illustrating the participants' level of technical understanding, concerns (privacy, cost, etc.), and technology preferences.

- The results section is mostly ‘phenomenal descriptive’ and contains many parts that are a direct summary of the participants’ statements.

- However, although the quotations are abundant, some themes are repeated without quotations, and there is a relative lack of analysis on the number and distribution of participants (country, gender, age, etc.).

- Interpretations of cultural differences are not consistently applied across the results section, and an analytical framework that takes cultural context into account is not explicitly stated.

- It is judged that there is a lack of theoretical explanation as to why this difference in attitude arose.

5. Discussion

- The discussion was well organized and summarized with key findings (lack of awareness, privacy concerns, affordability issues, etc.).

- The purpose and results of the study are logically explained by repeatedly emphasizing the message that ‘many participants recognize the potential of the technology, but practical barriers still exist.’

- The book's strong point is that it addresses the differences in perception between Pakistani and European older adults, and it was also impressive that it included some cultural explanations.

- Provides practical advice for a wide range of stakeholders, including designers, developers, policymakers, and family members.

- It is considered positive in terms of academic integrity that the researcher himself explicitly states limitations on the number of participants (small sample), digital accessibility bias, and limitations of online interviews.

- However, although there are some connections with existing literature, it is regrettable that most of them are merely repetitions of participant statements or observational interpretations.

- Compared to the implications of the ‘multicultural context’ addressed in this paper, it is judged that there is a lack of systematic structuring of the influence of culture.

- The practical suggestions are abundant, but the ‘theoretical contribution’ is unclear.

- There is a need to improve the general tone that appears despite the sample being biased towards the UK.

6. Conclusion

- The conclusion summarizes the key message drawn from the research results relatively well—that smart home technology has the potential to improve the quality of life (QoL) of older adults, but adoption is low due to barriers such as lack of awareness, privacy concerns, and cost burden.

- Throughout the paper, this study provides empirical insights based on qualitative exploration, but the conclusion does not explain whether any new theoretical implications or conceptual extensions have been made.

- The limitations of the discussion were clearly recognized, and the conclusion also mentions the need for follow-up research on various regional, cultural, and digital competency levels.

- The sentences are professional and logical, but they are somewhat redundant and lack a definitive message.

Reviewer #2: This study explored incentives and barriers to smart home technology adoption among older adults, examining how COVID-19 affected their quality of life. Through online interviews via Qualtrics with 21 participants aged 65–90 from various countries, the study identified that limited knowledge, affordability, privacy concerns, and integration issues hindered adoption, while perceived QoL improvements motivated interest.

Major edits:

1. The five smart home technology devices discussed are primarily manufactured in the EU or USA and mainly serve the US and European markets. Participants from Pakistan, particularly rural areas, may have limited familiarity with these devices as they are not readily available in their local markets.

2. Based on the first point, if participants are not familiar with these devices, their feedback on the influence of QoL is also limited, affecting the interpretation of their attitudes toward smart home devices.

Minor edits:

1. Include exclusion criteria for participant selection. A flowchart showing how many participants were included/excluded based on these criteria is preferred to clarify how participants were filtered or screened for eligibility.

2. Consider a paragraph or sentences of age subgroup to examine the influence of age on the perceived QoL impact of these smart home devices. For example, older adults over 80 may encounter more technical difficulties when using smart home devices in home settings.

.

Reviewer #1: No

Reviewer #2: **Yes:** Yijiong YangYijiong YangYijiong YangYijiong Yang

---

## [Author Response · Author response to Decision Letter 1]

9 Oct 2025

We have included a detailed point-by-point response to the reviewers’ comments in the “Files” section. Below, we also summarise the responses and the corresponding changes made to the manuscript.

Journal Requirements:

1. Thank you for pointing this out. We have carefully reviewed the PLOS ONE style templates and ensured that our manuscript conform to the journal’s formatting requirements.

2. Thank you for pointing this out. This project was part of Pireh Prizada’s PhD research. There is no specific funder listed under the University of St Andrews or St Leonard’s, so we have selected “no funder” since this project was not directly supported by a grant.

3. Thank you for this clarification. We confirm that this study was part of a PhD project funded by the University of St Andrews. The funders had no role in study design, data collection and analysis, decision to publish, or preparation of the manuscript.

4. We thank the editor for this important reminder. We have carefully reviewed the data within the manuscript and confirm that all personal information has been fully anonymised, in accordance with participant consent. We confirm that no hidden columns remain in the manuscript files.

5. We carefully considered the reviewer’s feedback. While no additional citation suggestions were provided, we have included new references to support the statements added in response to the reviewer’s comments. We sincerely appreciate the reviewer’s thoughtful guidance.

Reviewer #1:

Abstract - Page 1:

We sincerely thank the reviewer for the constructive feedback. We are pleased that the abstract was recognised for its clarity in emphasising the importance of studying smart home technology adoption among older adults, particularly regarding quality of life after COVID-19, and for outlining the study design and practical implications.

In response to the reviewer’s suggestions, we revised the abstract to:

Provide theme-focused highlights of the key findings rather than general statements.

Strengthen the results summary with clearer emphasis on major themes.

Restructure the abstract to enhance readability and ensure the core message stands out.

These changes are now reflected in red in the manuscript.

Introduction - Page 2, Lines: 20–47

We sincerely thank the reviewer for the detailed and insightful feedback on the Introduction. We appreciate the positive and constructive feedback regarding our manuscript.

In response to the reviewer’s suggestions, we made the following revisions:

To address this concern, we revised the text to more clearly identify the gaps our study addresses.

The final paragraph of the Introduction now better reflects prior cross-cultural research and clearly states the gap our study addresses.

Redundancies in the statement of research objectives were removed to avoid repetition.

Several sentences were restructured to improve readability, enhance clarity, and highlight the main message more directly.

Methods and Limitations:

We sincerely thank the reviewer for their careful evaluation and positive assessment of the methodological design.

In response to the reviewer’s constructive suggestions, we have made the following improvements:

To address concerns about sampling bias, we expanded the discussion of this limitation and added a more detailed explanation. These revisions can be found in the Limitation section, Page 21, Lines 762–781.

We strengthened the section by adding a revised paragraph which now provides clarity on the analysis procedures and reliability considerations. This can be found in the Methodology section, Page 3, Lines 104–109.

Results - Pages 8–18:

We sincerely thank the reviewer for the careful and thorough evaluation of the Results section.

In response to the reviewer’s constructive concerns, we have undertaken the following revisions:

Across all themes, we strengthened the analysis by linking findings to relevant theoretical frameworks and references, incorporating participant numbers/distribution. We also applied consistent cross-cultural comparisons (European vs. Pakistani participants). This can be found in the Results section from pages 8 to 18, highlighted in red throughout the section.

Tables 3 and 4 present the demographic distribution. Additional detail has now been provided in the Discussion section (Page 20, Lines 721–730), where the demographic breakdown is discussed.

Discussion - Pages 18–20:

We appreciate and thank the reviewer for the feedback on the Discussion section.

In response to the reviewer’s constructive concerns, we have made the following revisions:

We have strengthened the discussion section by embedding our findings more explicitly within existing empirical literature.

We have updated the discussion to make the cultural context on smart home technology adoption clearer between Pakistani and European participants.

We updated the section to connect our findings to established models of technology adoption and ageing, highlighted in red in the discussion section.

We have updated the section (Lines 721–724) to acknowledge our participants’ data being more skewed towards UK participants.

Conclusion - Pages 21-22:

We appreciate the reviewer for their feedback on the Conclusion section.

In response to the reviewer’s constructive suggestions, we have made the following improvements:

We have revised the conclusion and added modifications from Lines 793–797 and Lines 806–808 to make it more clear.

Reviewer 2:

Major Revisions:

1. Limitaions, Page: 21, Lines: 772–782

We thank the reviewer for this observation and constructive feedback.

In response, we have expanded the Limitations section to note this limitation and provided a mitigation strategy for future work.

2. Methods, Limitations

We thank the reviewer for raising this concern.

To address this point, we have extended our Limitation section to address this concern on Line 763–782, Page 21.

In addition, we have defined the strategy adapted for participants unfamiliar with the technologies during the interview, which has been included in the Methods section (Page 4, Lines 123–125).

Minor Revisions:

1. Methods, Eligibility, Page: 4, Lines: 131–136

We sincerely thank the reviewer for this valuable suggestion. In response, we have added a new section “Eligibility” which details inclusion and exclusion criteria. We have made it clear that all participants who signed up took part in the study.

2. Limitations, Page: 21, Lines: 763–768

We thank the reviewer for this suggestion. In response, we have expanded the Limitations section to include this.

---

## [Decision Letter · Decision Letter 1]

29 Oct 2025

Dear Dr. Pirzada,

Thank you for submitting your manuscript to PLOS ONE. After careful consideration, we feel that it has merit but does not fully meet PLOS ONE’s publication criteria as it currently stands. Therefore, we invite you to submit a revised version of the manuscript that addresses the points raised during the review process.

We look forward to receiving your revised manuscript.

Kind regards,

Dr Buna Bhandari

Academic Editor

PLOS ONE

Journal Requirements:

Reviewers' comments:

Reviewer's Responses to Questions

**Comments to the Author**

Reviewer #1: (No Response)

Reviewer #2: All comments have been addressed

2. Is the manuscript technically sound, and do the data support the conclusions?

Reviewer #1: (No Response)

Reviewer #2: Yes

3. Has the statistical analysis been performed appropriately and rigorously?

Reviewer #1: (No Response)

Reviewer #2: Yes

4. Have the authors made all data underlying the findings in their manuscript fully available?

Reviewer #1: (No Response)

Reviewer #2: Yes

5. Is the manuscript presented in an intelligible fashion and written in standard English?

Reviewer #1: (No Response)

Reviewer #2: Yes

Reviewer #1: - The researcher believes that many of the revisions made by the reviewers have been corrected.

- However, shortcomings are still being discovered.

- The specific details are as follows.

1. Further explanation of the theoretical background for TAM or UTAUT-based variables is needed.

2. Please clearly state the purpose of the study. (What are the determinants influencing older adults’ acceptance of smart homes post-COVID.

3. It would be a better study if we could compare the differences in perception across countries, gender, and age.

4. Please interpret the results in connection with prior research or theoretical variables that can support them in the discussion.

5. Please revise the entire thesis to have an academic structure rather than a report-style description (use thesis-style sentences and clear subparagraph topics).

Reviewer #2: The authors have thoroughly addressed the comments from the previous review. The paper can be considered for acceptance, provided no additional revisions are requested by other reviewers.

.

Reviewer #1: No

Reviewer #2: No

---

## [Author Response · Author response to Decision Letter 2]

19 Nov 2025

Journal Requirements:

We carefully considered the reviewer’s feedback. While no additional citation suggestions were provided, we have included new references to support the statements added in response to the reviewer’s comments. The following reference have been added to the paper:

Ishaq E, Bashir S, Zakariya R, Sarwar A. Technology Acceptance Behavior and

Feedback Loop: Exploring Reverse Causality of TAM in Post-COVID-19

Scenario. Frontiers in Psychology. 2021;12. doi:10.3389/fpsyg.2021.682507.

Mitzner TL, Savla J, Boot WR, Sharit J, Charness N, Czaja SJ, et al.

Technology Adoption by Older Adults: Findings From the PRISM Trial. The

Gerontologist. 2018;59(1):34–44. doi:10.1093/geront/gny113.

We thank the editor for this important reminder. All references have been checked for completeness and accuracy, and we confirm that no retracted papers have been included.

Reviewers 1 comments:

The researcher believes that many of the revisions made by the reviewers have been corrected.

- However, shortcomings are still being discovered.

- The specific details are as follows.

1. Further explanation of the theoretical background for TAM or UTAUT-based variables is needed.

Thank you for pointing this out. We expanded these explanations in the section where TAM (Page 10, Lines 299-303) or UTAUT (Page 19, Lines 664-668) first appear.

2. Please clearly state the purpose of the study. (What are the determinants influencing older adults’ acceptance of smart homes post-COVID.

We thank the reviewer for this helpful comment. To clearly state the purpose of the study, we have renamed the section from “Motivation and goal” to “Motivation and Objective” to improve clarity (Page 2, Line 48). We have also expanded the section discussing the impact of COVID-19 and added relevant references (Page 9, Line 228-233).

Citation Added: Ishaq E, Bashir S, Zakariya R, Sarwar A. Technology Acceptance Behavior and Feedback Loop: Exploring Reverse Causality of TAM in Post-COVID-19 Scenario. Frontiers in Psychology. 2021;12. doi:10.3389/fpsyg.2021.682507.

3. It would be a better study if we could compare the differences in perception across countries, gender, and age.

We thank the reviewer for this suggestion. A country-level comparison is presented in the Results (pp. 8–18). However, because only one participant was from Malta, we combined Malta with the UK under a broader ‘Europe’ category to allow a more meaningful comparison with Pakistan. We also appreciate the reviewer's interest in gender and age stratification. Consistent with standard gerontological practice, we define older adults as individuals aged 65 years and above, a threshold widely adopted in technology adoption research without further subdivision, as it reflects key life-stage transitions and ensures robust sample sizes for identifying broad patterns like digital literacy barriers. High-impact studies in this domain routinely analyse 65+ as a cohesive group to evaluate outcomes such as internet uptake and intervention efficacy (e.g., PRISM trial cited below and on page 3, Line 80-82). Therefore, stratification was not pursued here, as our objectives center on overarching trends rather than intra-group heterogeneity, and our data would yield underpowered subgroups. Gender was not a primary objective of the study, so we have acknowledged this in the limitations section (Page 21, Lines 796–799).

Reference Added: Mitzner TL, Savla J, Boot WR, Sharit J, Charness N, Czaja SJ, et al.

Technology Adoption by Older Adults: Findings From the PRISM Trial. The

Gerontologist. 2018;59(1):34–44. doi:10.1093/geront/gny113.

4. Please interpret the results in connection with prior research or theoretical variables that can support them in the discussion.

We thank the reviewer for the comment. We have expanded the discussion to make this clear. These revisions appear on Page 21, pages 756-764.

5. Please revise the entire thesis to have an academic structure rather than a report-style description (use thesis-style sentences and clear subparagraph topics).

Thank you for the comment. As the manuscript must strictly follow the PLOS ONE article format, we are unable to alter the structure.

---

## [Decision Letter · Decision Letter 2]

10 Dec 2025

Dear Dr. Pirzada,

Thank you for submitting your manuscript to PLOS ONE. After careful consideration, we feel that it has merit but does not fully meet PLOS ONE’s publication criteria as it currently stands. Therefore, we invite you to submit a revised version of the manuscript that addresses the points raised during the review process.

We look forward to receiving your revised manuscript.

Kind regards,

Dr Buna Bhandari

Academic Editor

PLOS One

Journal Requirements:

Reviewers' comments:

Reviewer's Responses to Questions

**Comments to the Author**

Reviewer #3: (No Response)

2. Is the manuscript technically sound, and do the data support the conclusions?

Reviewer #3: No

3. Has the statistical analysis been performed appropriately and rigorously?

Reviewer #3: N/A

4. Have the authors made all data underlying the findings in their manuscript fully available?

Reviewer #3: Yes

5. Is the manuscript presented in an intelligible fashion and written in standard English?

Reviewer #3: Yes

Reviewer #3: Motivation and objectives:

Why do you assume that older people has resistance towards smart homes? Recent research by Son Galanza et. al. suggests otherwise. It is important to avoid ageistic assumptions. Also the term elderly is considered stigmatising in gerontology, use older adults insteas.

Methodology:

Arguments for including people from age 65 is week and needs to be expanded. And why did you stop at age of 85?

I guess that the interviews were done in respondents native language - but that needs to be written out.

Why did not all participant receive an incentive (token of appreciation)?

Why include only one person from Malta? How is that context comparable to UK?

How were the age distributions per country? Are groups similar? What about other socio-economics such as income and education that may be of relevance for findings?

Where the three phases of interviews done at same time? The interviews seemed fairly short in time. Was sufficient depth reached?

Findings:

The results includes also other references - how come? I think that the findings from this study should only be reported. As it reads now findings lack depth. The structure with other references makes it hard to identify the contribution of this study - and very few quotes are used to support the findings.

Why use percentages to report findings when numbers are well below 100?

The figure does not make sense as the groups seem to differ much and group belonging is not reported.

Discussion:

TAM, UTAUT and DOI as theories are relevant - however rather mentioned than applied.

Overall:

The populations are rather diverse - while that may provide different perspectives I don't think the study reaches sufficient depth especially as the interviews had to be done online. For non-experience participants looking at pictures will not be sufficient to display adoption. And likely you are comparing two very/too different groups so I'm not sure that socio-cultural contexts and intergenerational dynamics make the difference (as written in the conclusion) - but rather that the Pakistani group had very little experience - while UK clearly had more.

.

Reviewer #3: No

---

## [Author Response · Author response to Decision Letter 3]

26 Jan 2026

Journal Requirements:

Response: We carefully considered the reviewer’s feedback. We have included new reference to support our statement added in response to the reviewer’s comments. The following reference have been added to the paper:

Galanza WS, Offerman J, Fristedt S, Iwarsson S, Malesevic N, Schmidt SM. Smart home technology to support engagement in everyday activities while ageing: A focus group study with current and future generations of older adults. PLOS ONE. 2025;20(1):1–20. doi:10.1371/journal.pone.0317352

Reviewers comments:

Reviewer 3, Motivation and objectives:

Why do you assume that older people has resistance towards smart homes? Recent research by Son Galanza et. al. suggests otherwise. It is important to avoid ageistic assumptions. Also the term elderly is considered stigmatising in gerontology, use older adults insteas.

Response: We thank the reviewer for this comment. Our findings align with the reviewer’s observation, which has now been made explicit and Galanza et al. (2025) is now cited to further support this in the manuscript (Line 67-69, Page 3).

The term older adults is used consistently throughout the manuscript. Elderly appears only when referring to stigmatising marketing language or when explicitly used by participants and the term is placed in quotation mark to make it clear (Line 257, page 9).

Reviewer 3, Methodology:

Arguments for including people from age 65 is week and needs to be expanded. And why did you stop at age of 85?

Response: All participants were adults aged 65 and above, age threshold chosen to be consistent with definitions of older adults in technology-adoption research [26] (This is already present on page 3, line 81-83).

No upper age limit was imposed. Participants aged 65–85 were included based on recruitment. The age range reflects the age of the oldest participant rather than an exclusion criterion, we have made this explicit in the Interviews section, page 3, Lines 89-90.

Reviewer 3, I guess that the interviews were done in respondents native language - but that needs to be written out.

Response: We have made this clearer in Methods section, Line 87-88, Page 3.

Reviewer 3, Why did not all participant receive an incentive (token of appreciation)?

Response: A small shopping voucher was provided to UK participants in accordance with local ethics approval; no incentives were offered in other contexts as it was not feasible within the approved logistical and administrative arrangements. We have made this clear on Line 134-136, page 4.

Reviewer 3, Why include only one person from Malta? How is that context comparable to UK?

Response: Participation was open internationally and based on voluntary recruitment; the single participant from Malta reflects uptake rather than targeted sampling. The analysis does not rely on country-level comparisons. This is mentioned already in Limitation section, page 21, line 804-806.

Reviewer 3, How were the age distributions per country? Are groups similar? What about other socio-economics such as income and education that may be of relevance for findings?

Response: The study was not designed to match age or socio-economic characteristics across countries. Income-related differences were explored qualitatively in the affordability analysis, while country-level or stratified comparisons were not intended; this is now clarified in the Limitations section, Page 21, Lines 806-808.

Reviewer 3, Where the three phases of interviews done at same time? The interviews seemed fairly short in time. Was sufficient depth reached?

Response: Yes, all three phases were conducted within a single interview session. Interviews lasted approximately 45 minutes on average, which allowed sufficient depth to explore participants’ experiences, perceptions, and adoption considerations across the three phases. We have made this clear on Line 156-157, Page 6.

Reviewer 3, Findings:

The results includes also other references - how come? I think that the findings from this study should only be reported. As it reads now findings lack depth. The structure with other references makes it hard to identify the contribution of this study - and very few quotes are used to support the findings.

Why use percentages to report findings when numbers are well below 100?

The figure does not make sense as the groups seem to differ much and group belonging is not reported.

Response: We thank the reviewer for the comments. References are included in the Findings section to contextualise our results within the existing literature, as requested in earlier review rounds and in line with common qualitative reporting practices. All findings reported are derived from the present study; external sources are used solely for interpretive context.

Both absolute numbers and percentages are provided to support transparency and aid interpretation given the sample size. Absolute numbers in addition to percentages were added following feedback from the previous round of reviews. The figure is intended to visually illustrate observed group differences, with explanations provided in the paper.

Reviewer 3, Discussion:

TAM, UTAUT and DOI as theories are relevant - however rather mentioned than applied.

Response: TAM, UTAUT, and DOI are referenced to situate the findings within established technology adoption literature rather than to serve as analytical frameworks. These were specifically added due to helpful comments form a previous round of reviews. The study is exploratory in nature and does not aim to apply or test these models.

Reviewer 3, Overall:

The populations are rather diverse while that may provide different perspectives I don't think the study reaches sufficient depth especially as the interviews had to be done online. For non-experience participants looking at pictures will not be sufficient to display adoption. And likely you are comparing two very/too different groups so I'm not sure that socio-cultural contexts and intergenerational dynamics make the difference (as written in the conclusion) but rather that the Pakistani group had very little experience, while UK clearly had more.

Response: We thank the reviewer for the feedback. The diversity of the sample was intentional and aligned with the study aims to capture breadth of perspectives. Interviews were in-depth (45 minutes), followed a consistent protocol, and adequately addressed the research objectives despite being conducted online.

Differences in experience levels across groups and other limitations are clearly acknowledged in the manuscript. The interpretation reflects contextual and socio-cultural factors rather than direct group comparison.

---

## [Editor Report · Decision Letter 3]

9 Mar 2026

Understanding older adults’ perception, acceptance, and adoption of smart home technologies∗

PONE-D-25-27595R3

Dear Dr. Pirzada,

We’re pleased to inform you that your manuscript has been judged scientifically suitable for publication and will be formally accepted for publication once it meets all outstanding technical requirements.

Kind regards,

Dr Buna Bhandari

Academic Editor

PLOS One

Additional Editor Comments (optional):

Please format the manuscript according to the PLOS ONE guidelines, including the required font size and line spacing throughout the document during proofreading.
---

## [Editor Report · Acceptance letter]

PONE-D-25-27595R3

PLOS One

Dear Dr. Wilde,

I'm pleased to inform you that your manuscript has been deemed suitable for publication in PLOS One. Congratulations! Your manuscript is now being handed over to our production team.

Kind regards,

on behalf of

Dr. Buna Bhandari

Academic Editor

PLOS One